# COVID-19 Vaccination Limits Systemic Danger Signals in SARS-CoV-2 Infected Patients

**DOI:** 10.3390/v14030565

**Published:** 2022-03-09

**Authors:** Roberta Angioni, Lolita Sasset, Chiara Cioccarelli, Ricardo Sánchez-Rodríguez, Nicole Bertoldi, Cristina C. Putaggio, Antonella Viola, Annamaria Cattelan, Barbara Molon

**Affiliations:** 1Fondazione Istituto di Ricerca Pediatrica—Città della Speranza, 35127 Padova, Italy; roberta.angioni@unipd.it (R.A.); chiara.cioccarelli@phd.unipd.it (C.C.); ricardo.sanchezrodriguez@unipd.it (R.S.-R.); bertoldi.nicole@gmail.com (N.B.); antonella.viola@unipd.it (A.V.); 2Infectious Disease Unit, Padova University Hospital, 35128 Padova, Italy; lolita.sasset@aopd.veneto.it (L.S.); cristina.putaggio@aopd.veneto.it (C.C.P.); annamaria.cattelan@aopd.veneto.it (A.C.); 3Department of Biomedical Sciences, University of Padova, 35131 Padova, Italy

**Keywords:** COVID-19, vaccination, DAMPs, personalized therapy

## Abstract

Vaccination with an mRNA COVID-19 vaccine determines not only a consistent reduction in the risk of SARS-CoV-2 infection but also contributes to disease attenuation in infected people. Of note, hyperinflammation and damage-associated molecular patterns (DAMPs) have been clearly associated with severe illness and poor prognosis in COVID-19 patients. In this report, we revealed a significant reduction in the levels of IL-1ß and DAMPs molecules, as S100A8 and High Mobility Group Protein B1 (HMGB1), in vaccinated patients as compared to non-vaccinated ones. COVID-19 vaccination indeed prevents severe clinical manifestations in patients and limits the release of systemic danger signals in SARS-CoV-2 infected people.

## 1. Introduction

Since the beginning of the SARS-CoV-2 pandemic, 359,561,272 cases have been confirmed as of 25 January 2021 and 5,635,677 deaths have occurred worldwide.

COVID-19 patients present with a variety of clinical manifestations, ranging from asymptomatic or mild respiratory illness to fulminant severe acute respiratory distress syndrome (ARDS) with extra-pulmonary complications [1].

Exploring this remarkable variability has been a main focus of COVID-19 research. Current evidence indicates that the immune response to the viral infection—depending on age, sex, viral load, genetics, and other known and unknown variables—largely defines the course of the disease [2]. Although the antiviral immune response is crucial for eliminating SARS-CoV-2, a robust and persistent antiviral immune response can also cause a massive production of inflammatory cytokines and damage to the host [3,4]. In addition, the overproduction of cytokines caused by an aberrant immune activation (termed a cytokine storm) may be a major cause of tissue damage [5,6]. Indeed, the cytokine storm can lead to apoptosis of epithelial cells and endothelial cells, and vascular leakage and, finally, result in ARDS, other severe syndromes, and even death [7,8]. Many studies have also demonstrated that T lymphocytes (T cell) (CD3+ CD4+ T cell and CD3+ CD8+ T cells) are reduced in COVID-19 and are significantly lower in SARS-CoV-2 severely ill patients in which high levels of C reactive protein (CRP) and IL-6 have been reported [9]. Vaccines represent the most effective means to prevent infectious diseases, and the SARS-CoV-2 vaccines in use, using different methods, mRNA, viral vector, show a good efficacy and safety profile [10,11,12]. While SARS-CoV-2 vaccines remain very effective on preventing severe disease and death they do not fully prevent transmission and infection. In addition, specific challenges in COVID-19 vaccination such as long-term immunity and avoiding cytokine storms need to be further explored. To date, there are poor data regarding the cytokine profile in SARS-CoV-2 fully vaccinated patients who subsequently acquired the infection. 

To address this issue, we measured the SARS-CoV-2-specific cytokine response in two groups of SARS-CoV-2 vaccinated and non-vaccinated patients who developed the disease in order to define the differentiating features of the inflammatory response and their association with severe disease.

## 2. Materials and Methods

### 2.1. Participants, Study Design, and Data Collection

All patients (aged > 18 years) who consecutively tested positive for SARS-CoV-2 at the Infectious and Tropical Diseases Institute of Padua, either as inpatient or outpatients, were included. All demographics and clinical characteristics were retrieved from medical health records. Testing for SARS-CoV-2 was performed by using real-time reverse transcriptase-polymerase chain reaction (RT-PCR) assay on nasopharyngeal swabs. Severity of COVID-19 was defined according to NIH definition (Asymptomatic: Individuals who test positive for SARS-CoV-2 but who have no symptoms that are consistent with COVID-19. Mild: individuals who have any of the various signs and symptoms of COVID-19 but who do not have shortness of breath, dyspnea, or abnormal chest imaging. Moderate: individuals who show evidence of lower respiratory disease during clinical assessment or imaging and who have an oxygen saturation (SpO_2_) ≥ 94% on room air at sea level [13]. Severe: Individuals who have SpO_2_ < 94% on room air at sea level, a ratio of arterial partial pressure of oxygen to fraction of inspired oxygen (PaO_2_/FiO_2_) < 300 mm Hg, a respiratory rate > 30 breaths/min, or lung infiltrates > 50%. Critical Illness: Individuals who have respiratory failure, septic shock, and/or multiple organ dysfunction). For each patient who agreed to participate by consenting, blood samples were collected and stored to dose immunological parameters as per our objectives. Time from both positive results for SARS-CoV-2 and symptom onset and blood sampling was recorded. The local ethics committee was notified about the study protocol. The study was performed according to the ethical guidelines of the Declaration of Helsinki (7th revision). All the patients gave their written informed consent and all analyses were carried out on anonymized data as required by the Italian Data Protection Code (Legislative Decree 196/2003) and the general authorization issued by the Data Protection Authority.

### 2.2. Luminex and ELISA Assay

Peripheral blood from enrolled COVID-19 inpatients was collected in EDTA tubes and stored at 4 °C prior to processing. Plasma was isolated by Ficoll procedure and stored at −80 °C until the analysis. Some 48 analytes were measured by multiplex biomarker assays, based on Luminex xMAP technology (Merck Millipore, Burlington, MA, USA) following manufacturer’s instructions. Plasma DAMPs (S100A8 and HMGB1) were evaluated by ELISA (antibodies-online GmbH, Aachen, Germany) according to the manufacturer’s instructions.

### 2.3. Statistical Analysis

Data were analyzed using the Prism Software (GraphPad, La Jolla, CA, USA). Statistical comparison between the two groups was carried out using unpaired nonparametric Mann–Whitney. Differences were considered statistically significant at confidence levels * *p* < 0.05 or ** *p* < 0.01. Data plotted were expressed as mean with standard error of mean (SEM).

## 3. Results

Between 12 August and 2 September 2021, 47 patients tested positive in our setting. The baseline characteristics of the studied population, by vaccination status (non-vaccinated = group A and vaccinated = group B) are depicted in Table 1. Two patients were excluded: one for an ongoing pulmonary tuberculosis and one for long COVID-19, probably due to an overlapping hematological disease. Therefore, 45 patients were considered in our analysis (Table 1). Overall, 29/45 (63%) were males and the median age was 61 (IQR: 48–79) years. Most of the subjects (39/45, 86.6%) were admitted due to severity of COVID-19, while six were managed as outpatients. More than 50% of the patients (23/45, 51.1%) had not received any vaccine against SARS-CoV-2 (NoVax), while 49.9% of the subjects had received at least one dose before the infection (Vax) (Figure 1A). The median age in the NoVax group was significantly lower than that in the Vax group (57 years, IQR: 41.5–62 vs. 79 years, IQR: 49.5–87, *p* < 0.05). The male gender was equally represented in both groups (Figure 1B). 

In terms of comorbidities, the most common overall were: malignancies (11/45, 24.4%), diabetes (10/45, 22.2%), ischemic heart disease (6/45, 13.3%), chronic obstructive pulmonary disease (COPD) (4/45, 8.9%). The two groups were not significantly different in terms of comorbidities (Table 1). Patients who were managed in the outpatient’s setting all had mild disease. Among those who were admitted (39): 1 (2.5%) had moderate disease, 33 (84.7%) severe disease and 5 (12.8%) critical disease requiring intensive care unit admission (Figure 1C). One patient, a 92-year-old woman with multiple comorbidities, died. 

Most of the patients 37/46 (80.4%) who were hospitalized received oxygen support. The proportion of patients who required oxygen support was significantly different between the two groups. A total of 25/45 patients needed low flow oxygen; among them, 14 (60.9%) were in the NoVax group and 11 (50%) in the Vax group (*p* value 0.333), 7/45 patients needed high flow oxygen; among them, 6 (26,1%) were in the NoVax group and 1 (4.5%) in the Vax group; *p* value 0.054). Of the patients, 5/45 required mechanical ventilation, 2 (9.1%) in the NoVax group and 3 (13%) in theVax group (*p* value 0.522) (Table 1).

Current evidence indicates that the cytokine storm plays a crucial role in determining COVID-19 severe outcomes [14]. To analyze the cytokine profile of our patients, we measured the circulating levels of 48 cytokines in the plasma of both Vax and NoVax SARS-CoV-2 infected patients, stratified according to patient age (Appendix A). Among all the analytes, we pointed out a significant difference in the levels of IL-1ß between the two groups (Figure 1D). Intriguingly, the plasmatic IL-1ß content positively correlated with higher hospitalization time, age and disease severity [15,16]. 

IL-1ß is a cytokine released through the activation of the inflammasome, a multimeric complex triggered by pathogen-associated (PAMPs) and/or damage-associated (DAMPs) molecular patterns. Of note, recent reports clearly indicated a strong correlation between DAMP release and poor clinical outcomes for COVID-19 patients [17,18]. We found that vaccination is associated to a relevant reduction of the levels of S100A8 (Figure 1E) and HMGB1 (Figure 1F). Although the difference was significant for HMGB1, we only observed a trend of decrease for S100A8 that can be explained by a different pattern of expression of the DAMPS in our cohort. These DAMPs have been already detected in the serum of COVID-19 patients, where they strongly correlate with higher risk of ICU admission and death [14]. 

Given the systemic inflammatory status of COVID-19 patients, we evaluated liver functionality in our cohort. The level of both alanine (ALT) and aspartate (AST) were comparable in the two groups. On the other hand, we found high levels of circulating glutamate dehydrogenase activity (GDH) in the NoVax group (Appendix A), possibly suggesting mitochondrial damage in these patients. In this line, high circulating mitochondrial DNA has been already defined as a potential early indicator for poor COVID-19 prognosis [19].

## 4. Discussion

Overall, our results indicate that COVID-19 vaccination prevents the release of systemic danger signals and IL-1ß in SARS-CoV-2 infected patients. Moreover, our data provide a new insight into the definition of the proper therapeutic paths for non-vaccinated patients as the direct targeting of IL-1ß.

Collectively, we confirm that vaccination represents the best strategy to prevent potential long-term side-effects caused by the SARS-CoV-2 related inflammation.

## Figures and Tables

**Figure 1 viruses-14-00565-f001:**
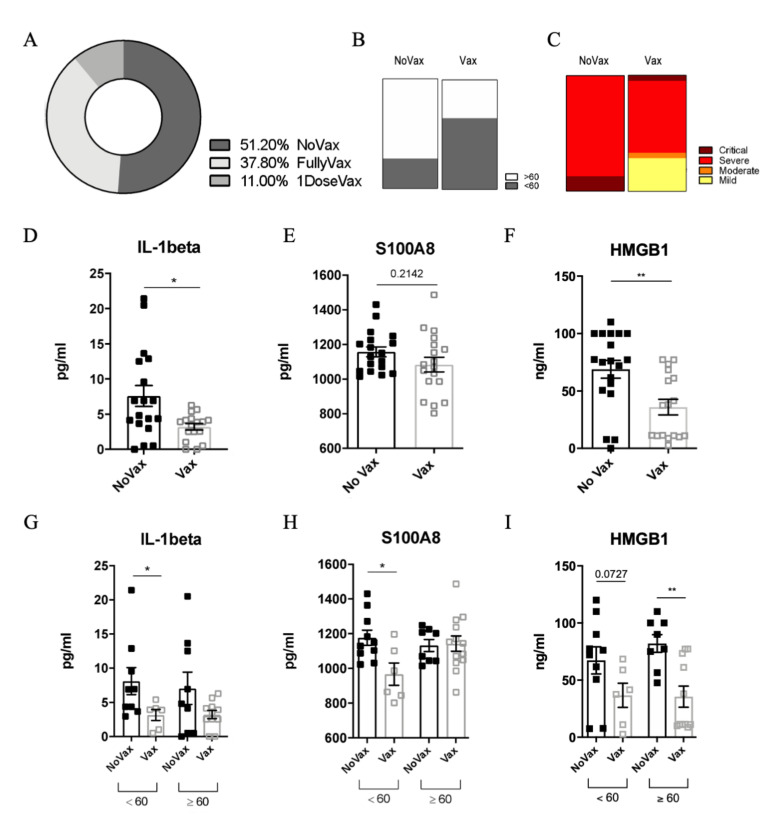
(**A**) Percentage of non-vaccinated (NoVax), vaccinated (Vax) and 1 dose vaccinated patients in our cohort. Age (**B**) and disease severity (**C**) distribution in NoVax and Vax patients. IL-1ß (**D**), S100A8 (**E**), and HMGB1 (**F**) plasma concentration (pg/mL) in NoVax and Vax patients. IL-1ß (**G**), S100A8 (**H**), and HMGB1 (**I**) plasma concentration (pg/mL) in NoVax and Vax patients stratified by age (younger or older than 60 years old). Differences were considered statistically significant at confidence levels * *p*  <  0.05 or ** *p*  <  0.01.

**Table 1 viruses-14-00565-t001:** Demographic and clinical data of enrolled COVID-19 patients.

	Vax (*n* = 22)	NoVax (*n* = 23)	*p* Value
**Demographics**			
Age, year, median	79	57	0.002
Male (%)	15 (68.2%)	14 (60.9%)	0.421
Comorbidities			
None (%)	9 (40.9)	13 (56.5)	0.179
1–3 (%)	13 (49.1)	10 (43.5)	0.566
>3 (%)	2 (9.09)	6 (26.1)	0.107
**Severity of COVID-19**			
Mild	6 (27.3%)	0	0.009
Moderate	1 (4.5%)	0	0.489
Severe	13 (59.1%)	20 (87%)	0.037
Critical	2 (9.1%)	3 (13%)	0.522
**Supplemental oxygen therapy**			
None	8 (36.4%)	0 (0.0%)	0.001
Low flow	11 (50%)	14 (60.9%)	0.333
High flow	1 (4.5%)	6 (26.1%)	0.054
Mechanical ventilation	2 (9.1%)	3 (13%)	0.522

## Data Availability

Not applicable.

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
