# Peer review of "COVID-19 Vaccination Limits Systemic Danger Signals in SARS-CoV-2 Infected Patients"

_viruses, 2022, doi:10.3390/v14030565_

Round 1
Reviewer 1 Report
The authors have analyzed inflammatory cytokine (IL-1 ß) and DAMPs markers (S100A8 and HMGB1) in a cohort of COVID-19 hospitalized patients. This brief report demonstrates a higher level of such inflammatory markers in non-vaccinated patients in respect to vaccinated patients.
The manuscript is interesting however I have the following suggestions.
1) Please correct "IL-1 ß" in the manuscript (e.g. line 16).
2) Please correct "SARS-CoV-2" in the manuscript (e.g. lines 43 and 47).
3) Please add statistical analysis information in the Materials and Methods section.
4) Line 69, please add the following reference: https://www.covid19treatmentguidelines.nih.gov/overview/clinical-spectrum/
5) Figure 1B and 1C are not clear and should be corrected.
6) References are limited: please add more citations.
7) How are correlated the analyzed markers with COVID-19 severity?
8) Line 81: "48 analytes were measured by multiplex biomarker assays". Why such measurements are not included in this report?
Author Response
We thank the Reviewer for the constructive comments. Please consider our point-by-point reply to the reviewer’s comments.
The authors have analyzed inflammatory cytokine (IL-1 ß) and DAMPs markers (S100A8 and HMGB1) in a cohort of COVID-19 hospitalized patients. This brief report demonstrates a higher level of such inflammatory markers in non-vaccinated patients in respect to vaccinated patients.
The manuscript is interesting however I have the following suggestions.
- Please correct "IL-1 ß" in the manuscript (e.g. line 16).
We thank for the comment. We changed the text accordingly.
- Please correct "SARS-CoV-2" in the manuscript (e.g. lines 43 and 47).
We thank for the comment. We changed the text accordingly.
- Please add statistical analysis information in the Materials and Methods section.
We apologies for the mistake. We added the paragraph “2.3. “Statistical analysis” to the material and methods section.
- Line 69, please add the following reference: https://www.covid19treatmentguidelines.nih.gov/overview/clinical-spectrum/
We thank for the comment. We changed the text accordingly.
- Figure 1B and 1C are not clear and should be corrected.
We thank for the comment. We improved the quality of the Figure1. However it is possible that the low resolution of images can be due to the PDF-conversion of the .text file. We also attached the .tif files.
- References are limited: please add more citations.
We thank for the comment. We added the following references, as indicated in the text:
- Zhou, X. & Ye, Q. Cellular Immune Response to COVID-19 and Potential Immune Modulators. Front Immunol 12, 646333 (2021).
- Merad, M. & Martin, J. C. Pathological inflammation in patients with COVID-19: a key role for monocytes and macrophages. Nat Rev Immunol 20, 355–362 (2020).
- Sinha, P., Matthay, M. A. & Calfee, C. S. Is a “Cytokine Storm” Relevant to COVID-19? JAMA Internal Medicine 180, 1152–1154 (2020).
- Bonaventura, A. et al. Endothelial dysfunction and immunothrombosis as key pathogenic mechanisms in COVID-19. Nat Rev Immunol 21, 319–329 (2021).
- How are correlated the analyzed markers with COVID-19 severity?
We thank for the comment. We highlighted in the text (lines 136-137) the correlations between DAMPS and COVID-19 severity. As for our cohort we have no Mild/moderate patients in the non-Vaccinated group, thus we cannot properly address this point in our cohort.
- Line 81: "48 analytes were measured by multiplex biomarker assays". Why such measurements are not included in this report?
We thank for the comment. We included the 48 analytes measurement in the supplementary figures.
Reviewer 2 Report
Although this study had several limitations, clearly stated by the authors, however the work covers an item under-represented in literature, and may serve as a stimulus for other groups to replicate the study
Author Response
We really thank the Reviewer for the positive comments on our manuscript.
Reviewer 3 Report
This is a simple and straightforward study revealing the reduction of DAMPs in vaccinated patients. The subjects number is relatively small, but no clear overclaim was found. I have only few suggestions for the authors:
Line 125: Figure 1E, the reduction of S100A8 is slight and not significant. The authors should state this more clearly and be better to dicuss it.
Figure 1C: the figure is blurry in manuscript for reviewing.
Line 33: a citation is needed here.
Line 35: "disease tissue" is suggested to be replaced by another expression.
Author Response
We thank the Reviewer for the constructive comments. Please consider our point-by-point reply to the reviewer’s comments.
This is a simple and straightforward study revealing the reduction of DAMPs in vaccinated patients. The subjects number is relatively small, but no clear overclaim was found. I have only few suggestions for the authors:
Line 125: Figure 1E, the reduction of S100A8 is slight and not significant. The authors should state this more clearly and be better to dicuss it.
We thank for the comment. We implemented the text accordingly (lines 139-141).
Figure 1C: the figure is blurry in manuscript for reviewing.
We thank for the comment. We improved the quality of the Figure1. However it is possible that the low resolution of images can be due to the PDF-conversion of the .text file. We also attached the .tif files.
Line 33: a citation is needed here.
We thank for the comment. We added the following reference:
- Yang, L. et al. The signal pathways and treatment of cytokine storm in COVID-19. Sig Transduct Target Ther 6, 1–20 (2021).
- Merad, M. & Martin, J. C. Pathological inflammation in patients with COVID-19: a key role for monocytes and macrophages. Nat Rev Immunol 20, 355–362 (2020).
Line 35: "disease tissue" is suggested to be replaced by another expression.
We thank for the comment. We changed the text accordingly.
Reviewer 4 Report
The article proposed by Angioni et al constitutes a short additional note on the interest of vaccination against COVID-19. By recalling the limits of this vaccination, the authors show, thanks to a cohort of 45 patients vaccinated or not that vaccination, contributes to significantly reduce the inflammatory response associated with infection by SARS-CoV2, with the consequence of a much better outcome for vaccinated patients. This study provides additional results to those already collected on the benefits of vaccination. The study protocol is rigorous with particular care to reduce as much as possible the biases that are inevitable for this type of study.
In the end, I recommend the publication of this article.
Author Response

(The authors gave the same response as above.)

Round 2
Reviewer 1 Report
The authors have partially responded to the reviewer's suggestions.
There are some criticisms that need to be addressed:
- Line 69, please add the following reference: https://www.covid19treatmentguidelines.nih.gov/overview/clinical-spectrum/
- Figure 1B: both columns are labeled "Vax".
Author Response
We thank the Reviewer for the suggestions.
The indicated Reference has been included at line 69 (ref 13).
Figure ha been modified accordingly.